# Attosecond spectroscopy of molecular charge transfer uncovers a 1.5-fs delay in population transfer

Danylo T. Matselyukh [1,7], Florian Rott [2,7], Thomas Schnappinger [2,3] ✉, Pengju Zhang [1,4], Zheng Li [5,6], Jeremy O. Richardson [1], Regina de Vivie-Riedle[2] & Hans Jakob Wörner [1] ✉

The transfer of population between two intersecting quantum states is the most fundamental event in many dynamical processes in physics, chemistry, biology, and material science. Any two-state description of such processes requires population leaving one state to instantaneously appear in the other. We show that coupling to additional states, present in all real-world systems, can cause a measurable delay in population transfer. Using attosecond spectroscopy supported by quantum-chemical calculations, we measure a delay of $1.46 \pm 0.41$ fs at a charge-transfer crossing in $CF_3I^+$, where an electron hole moves from the fluorine atoms to iodine. Our measurements also resolve the other fundamental quantum-dynamical processes involved in the charge-transfer reaction: a vibrational rearrangement time of $9.38 \pm 0.21$ fs (during which the vibrational wave packet travels to the state crossing) and a population-transfer time of 2.3–2.4 fs. Our work shows that delays in population transfer readily appear in otherwise-adiabatic reactions and predicts them to be on the order of a single-femtosecond for molecular valence-state crossings. These results have implications for many research areas, such as atomic and molecular physics, charge transfer, or light harvesting.

The energetic crossing of quantum states is a general problem, critical for our understanding of most quantum systems' dynamics. In physics, relevant systems include nuclear spins in varying magnetic fields[1] (directly applicable to quantum information and computing), the dynamics of atoms and molecules in external electric or magnetic fields, particularly Stark-coupled Rydberg states[2], and the treatment of quantum decoherence and noise[3], while in chemistry these include charge transfer (CT)[4,5], the passage through avoided crossings and conical intersections[6].

Due to this ubiquity, the quantum-mechanical treatment of state crossings dates back to the advent of quantum mechanics with the Landau–Zener–Stueckelberg–Majorana (LZSM) transition[7]. Extensions of this work have increased the number of states that cross at a single point (bow-tie crossing[8]), or allow the treatment of successive crossings (equal-slope[9]), but studies of second-order interactions at state crossings (i.e., the coupling of two crossing states through a third) are scarce[10,11].

Moreover, these crossings have been generally treated with a focus on determining the ultimate probability of diabatic/adiabatic population transfer[7] or the overall rate[12]. Attosecond science opens an entirely new perspective on the dynamics of such crossings, allowing not only the final populations to be observed, but also the transient dynamical processes governing their evolutions.

[1]Department of Chemistry and Applied Biosciences, ETH Zürich, Zürich, Switzerland. [2]Department of Chemistry, LMU Munich, Munich, Germany. [3]Department of Physics, Stockholm University, Stockholm, Sweden. [4]Beijing National Laboratory for Condensed Matter Physics, IOP CAS, Beijing, People's Republic of China. [5]School of Physics, Peking University, Beijing, People's Republic of China. [6]Collaborative Innovation Center of Extreme Optics, Shanxi University, Taiyuan, Shanxi, China. [7]These authors contributed equally: Danylo T. Matselyukh, Florian Rott. ✉e-mail: thomas.schnappinger@fysik.su.se; hwoerner@ethz.ch

Applying such a time-resolved strategy to molecular CT, this study identifies previously-unobserved transient experimental observables that offer unique insights into the nature of state crossings. We show, both experimentally and theoretically, that even in the case of a direct two-state crossing of the type treated in the LZSM problem[7] or CT with Marcus theory[13], additional intermediate states can influence the transient dynamics, leading to a unique observable: a delay in population transfer. This observable can reveal the participation of previously neglected intermediate states in the quantum-state crossings of multi-level systems. Accordingly, these results have implications for investigations of the mechanism (superexchange vs injection) driving CT along molecular bridges[14] or even the study of superconductivity, where the formation of Cooper pairs can be modulated by the superexchange interaction[15].

For molecular valence-state crossings, we predict these delays in population transfer to be on the order of a single femtosecond. We experimentally confirm and quantify an example of such a delay by studying the sub-femtosecond dynamics of the $\tilde{B}$ state of the trifluoromethyliodide cation ($CF_3I^+$), which undergoes considerable electron reorganization and CT between photoionization and dissociation[16,17]. These CT dynamics are probed by extreme ultraviolet (XUV) attosecond transient-absorption spectroscopy (ATAS), offering sub-femtosecond time resolution and indirect spatial sensitivity through its element specificity. ATAS is coupled with in-situ mass spectrometry, serving as an independent probe of the photoionization/dissociation, and ab-initio multiconfigurational second-order perturbation theory restricted active space (RASPT2) calculations, state-of-the-art calculations capable of accurately treating the spin-orbit coupling (SOC) in both the valence- and core-ionized states of $CF_3I^+$.

Our measurements reveal that strong-field ionization (SFI) with a carrier-envelope-phase (CEP) stable few-cycle pulse prepares a vibrational wave packet in the $\tilde{B}$ state, which reaches a state crossing in $9.38 \pm 0.21$ fs, the electronic-state population leaves (appears in) the $\tilde{B}$ ($\tilde{E}$) state with a timescale of $2.44 \pm 0.44$ fs ($2.34 \pm 0.36$ fs). In agreement with our theoretical model, we observe a non-instantaneous transfer of the population—the appearance of the electron hole on the iodine atom (corresponding to the $\tilde{E}$ state) is delayed by $1.46 \pm 0.41$ fs with respect to its disappearance from the fluorine lone-pair orbitals (corresponding to the $\tilde{B}$ state). Apart from revealing population transfer delay, our work represents the first time that the coupled electronic-nuclear processes of CT have been directly resolved and deconvolved. Previous measurements were able to estimate the overall timescale of state crossing/CT through exponential fits of experimental observables (e.g., yielding a sub-7-fs timescale for internal conversion in $C_2H_4^+$[18]) or by assuming kinetic rate laws (e.g., in core-hole clock experiments[19]). Here, owing to an instrument-response function of $\sigma_{Inst} = 1.00 \pm 0.05$ fs (see Fig. S2), we have been able to go beyond the traditional assumption of exponential kinetics.

## Results and discussion

To investigate the effect that a third (intermediate) state has on the dynamics of a two-state crossing we introduce the following model Hamiltonian, which depends on a reaction coordinate $x$:

$$\hat{H} = \hat{K}(\hat{p}) + \hat{V}(x) = \left( \frac{\hat{p}^2}{2m} - \alpha_0 x \right) \mathbf{I} + \begin{bmatrix} 0 & \gamma & 0 \\ \gamma^* & c - \alpha x & \gamma \\ 0 & \gamma^* & -2\alpha x \end{bmatrix} \quad (1)$$

where $\hat{K}, \hat{V}, \hat{p}$ are the kinetic-energy, potential-energy, and momentum operators, respectively, $m$ is the mass, $\alpha_0$ is the gradient of the initial state, $\alpha$ is the gradient difference between the intermediate state and the crossing states, and $c$ and $\gamma$ are the energy offset and diabatic coupling (using atomic units throughout), respectively. The basis of diabatic electronic states consists of the initial $|0\rangle$, intermediate $|1\rangle$,

and final $|2\rangle$ states; a diabatic crossing between $|0\rangle$ and $|2\rangle$ occurs at $x = 0$. Although the direct coupling between the initial and final states is set to zero, we find that the presence of a small coupling does not fundamentally modify the results.

We choose the system mass to be the reduced mass of the $CF_3$ and I moieties, and the Hamiltonian parameters to yield potential energy curves (PECs) representative of typical molecular valence states. The PECs for $\alpha_0 = 0.1$, $\alpha = 0.2$, $\gamma = 0.065$ and $c = 0.114$ are plotted in Fig. 1A, B. We investigate the temporal dynamics of this system by solving the TDSE using the split-operator method[20] (see Section S1.2). Initializing the system (as a Gaussian centered at $x = -0.4 = x_0$ with a probability width at half maximum of 0.192 and zero momentum) in the adiabatic ground state uphill of the crossing, results in the wavepacket passing through the crossing and transferring population from $|0\rangle$ to $|2\rangle$. The transient population dynamics can be found in Fig. 1C. To investigate these systematically, they are fit with cumulative distribution functions (CDFs) of asymmetric generalized normal distributions (AGNDs)[21] whose parameters are the location $\mu$, the scale $\sigma$, and the shape $\kappa$. $\mu$ specifies the median of the distribution and $\kappa$ the degree of skewness. As a result, these functions can capture the temporally-asymmetric nature of the rate of population change, which LZSM-like transitions are well known for. Section S1.1 discusses AGNDs and LZMS transitions in more detail.

For the system parameters presented in Fig. 1A–C, the population dynamics exhibit an important transient feature: a delay between the loss of population in $|0\rangle$ (light-blue) and gain in $|2\rangle$ (purple) due to it flowing through $|1\rangle$ (yellow). This delay is extracted from the difference of the position parameters $\mu$ of the fits. It is emphasized visually by inverting and rescaling the fit of the population in $|0\rangle$ such that it matches the asymptotic populations of the fit of the population in $|2\rangle$. This 'renormalization' of the fits shall be used throughout this work to visualize the population-transfer delay.

Figure 1D investigates how the population dynamics change as a function of $\gamma$ and $c$. We find that they can be separated qualitatively into four regimes (divided with black dashed lines and illustrated with inset schematics). When the offset $c$ is large (two upper quadrants), the delay is negligible and the system behaves as if it were made up of only two states. This can be explained by applying the Schrieffer–Wolff transform[22] (see Section S1.3), which removes the coupling to the intermediate state to lowest order. As a result, the Hamiltonian can then be re-written as a regular two-state crossing, whose behavior is well understood:

$$\hat{V} \rightarrow \hat{V}_{SW} = -\alpha_0 x \mathbf{I} + \begin{bmatrix} \frac{2\gamma^2}{\alpha x - c} & 0 & \frac{2c\gamma^2}{\alpha^2 x^2 - c^2} \\ 0 & c - \alpha x - \frac{4\gamma^2 c}{\alpha^2 x^2 - c^2} & 0 \\ \frac{2c\gamma^2}{\alpha^2 x^2 - c^2} & 0 & -2\alpha x - \frac{2\gamma^2}{\alpha x + c} \end{bmatrix}. \quad (2)$$

For large $c$, the dynamics are primarily governed by the effective coupling $\gamma_{SW} = \frac{2\gamma^2}{c}$, which determines whether the state crossing occurs diabatically (low transition probability, top-left quadrant) or adiabatically (high transition probability, top-right quadrant). In both cases, the population of $|1\rangle$ remains negligible, no delay in population transfer is present, i.e., the fits return the same location parameters $\mu_{|0\rangle} = \mu_{|2\rangle}$, and the same shape parameters $\kappa_{|0\rangle} = \kappa_{|2\rangle}$. In the case of CT, this regime corresponds to bridge-mediated superexchange[14].

When $c$ is lowered (bottom-right quadrant) and becomes comparable to the wavepacket's kinetic energy at $x = 0$ ($K_{x=0} \approx -\alpha_0 x_0$), it becomes possible to transfer population into $|1\rangle$ and the Schrieffer–Wolff transform fails. This leads to a unique and robust observable: a measurable delay in the population transfer $\mu_{|0\rangle} < \mu_{|2\rangle}$. In the case of bridge-mediated CT, this regime corresponds to injection[14]. The bottom-left quadrant shows an unphysical negative delay due to a second state crossing between $|0\rangle$ and $|1\rangle$ at $x > 0$. Such dynamics fall under the category of triangular crossings[23], in which wavepackets

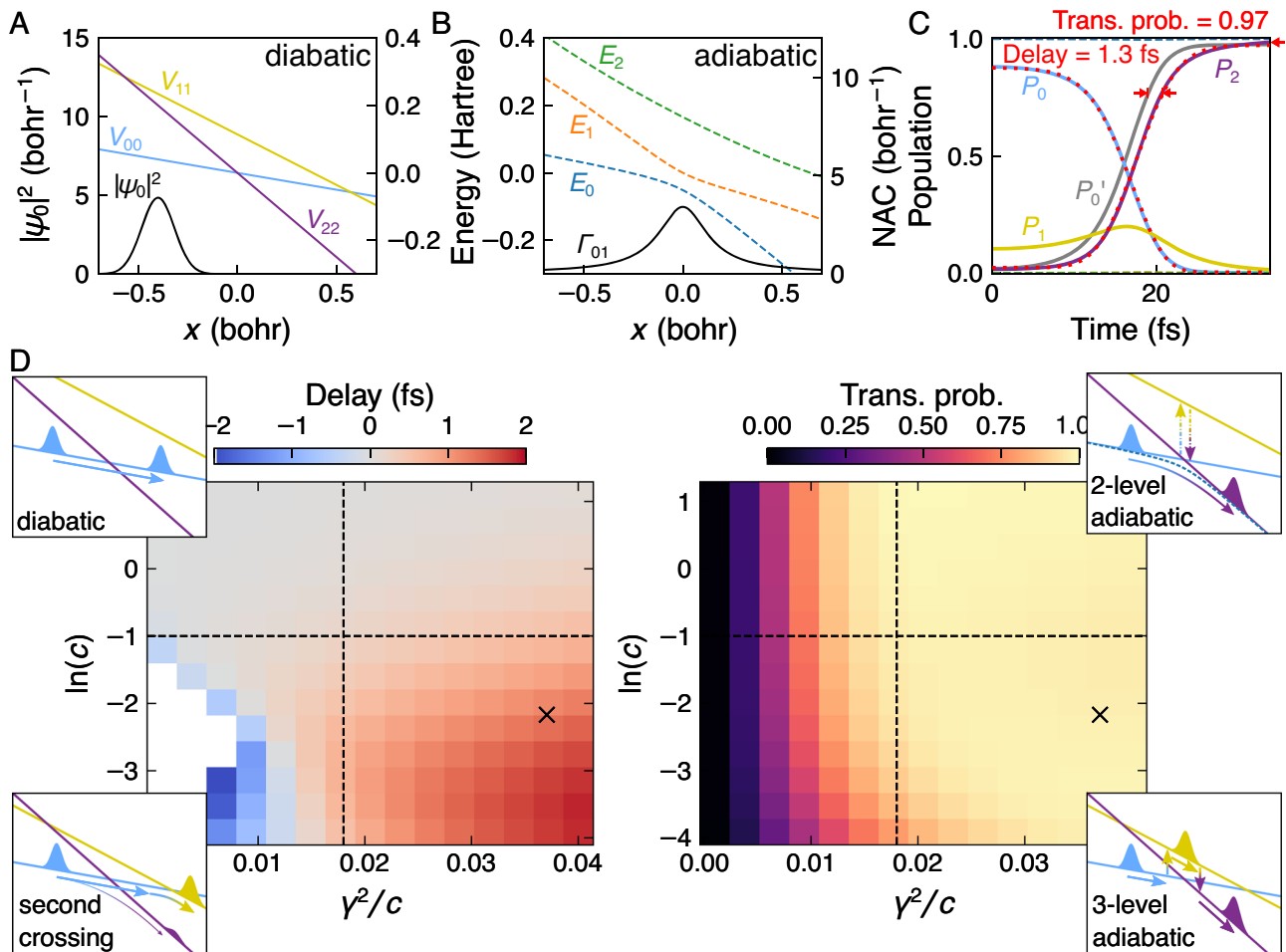

**Fig. 1 | Origin and magnitude of delays in population transfer. A** A three-state crossing in the diabatic basis; the diagonal entries of $\hat{V}(x)$ are plotted alongside the initial wavefunction $\psi_0$ (black). **B** The adiabatic PECs $E(x)$ (colored) reveal the avoided crossing between the bottom states. The associated non-adiabatic coupling (NAC) $\Gamma_{01}$ is also plotted (black). **C** The time-dependent populations resulting from a TDSE propagation of $\psi_0$ according to $\hat{H}(x)$ are shown. Solid (dashed) lines show the diabatic (adiabatic) populations. $P_0$ (light-blue) and $P_2$ (purple) are fit with AGND CDFs (red dotted). To emphasize the 1.3 fs delay in population transfer (red double-arrow), an amplitude-inverted and rescaled replica of the $P_0$ AGND fit is plotted (gray). The diabatic population transfer probability (taken to be the maximum $|2\rangle$ population over the 29.0 fs-long simulation) is indicated with a single red arrow. **D** A parameter scan revealing the dependence of the delay and population transfer probability on the energy spacing $c$ and the Schrieffer–Wolf two-state coupling $\gamma_{SW}/2 = \gamma^2/c$ (see Eq. 2). Four distinct regions are identified and illustrated with inset schematic diagrams. A black cross marks the parameters of the simulation shown in (**A**–**C**).

from two different crossings interfere at a third, and are therefore irrelevant to our discussion of single point multi-state crossings. As a result, and thanks to its convergence to an effective 2-state system at large $c$, our model can be verified by simply observing a population transfer delay.

To experimentally observe this delay, we study the temporal evolution of the $\tilde{B}$ state of $CF_3I^+$. The electronic structure and PECs of $CF_3I^+$ (presented in Fig. 2) are investigated with the help of RASPT2 ab-initio calculations (introduced below, details in Section S3): Fig. 2A presents the resulting diabatic PECs along with the dissociation limits of each state, while Fig. 2B presents their leading electronic configuration, i.e., character. The calculations reveal the presence of a CT crossing between the $\tilde{B}$ $^2A_2$ state (hole in the $lp_3$ fluorine lone-pair orbitals) and $\tilde{E}$ $^2A_1$ state (2-hole-1-particle configuration with a double hole in $lp_1$ and $lp_2$ and an electron in $\sigma_5^*$) very close to the Frank-Condon (FC) region (wavy yellow arrow). A transition between these states requires the rearrangement of two electrons (see Fig. 2B), suggesting that it cannot occur instantaneously and that an intermediate electronic state $\tilde{I}$ of mixed $(lp_3)^1(\sigma_4)^2(lp_{1,2})^3(\sigma_5^*)^1$ character might be involved. We confirm the existence of such a state ~ 4 eV above the CT crossing using the density functional theory/multireference configuration interaction (DFT/MRCI) method[24–26], a more computationally

efficient multireference approach (see Sections S3.2 and S3.5 for details).

Our attosecond experimental methods are illustrated in Fig. 3A–C and discussed in the "Methods" section. A CEP-stable 5.2-fs intense optical pump pulse is used to strong-field ionize $CF_3I$, while a time-delayed isolated attosecond pulse probes the ensuing dynamics through iodine-4d core-to-valence transitions (between 45 and 55 eV). Fluctuations in the attosecond probe pulse's spectrum were compensated using singular-value-decomposition-based filtering[27,28] (see Section S2.3). The temporal confinement of SFI[29,30] results in a measured experimental cross-correlation of $\sigma_{Inst} = 1.00 \pm 0.05$ fs (Fig. S2). SFI simultaneously prepares multiple electronic states of the cation, initiating different vibrational dynamics in each. Note that the quasi-static (i.e., electronically-adiabatic) nature of SFI in small molecules[31] is the reason why we initialize all of our three-state simulations in the adiabatic ground state.

The dynamics in different electronic states of $CF_3I^+$ are deconvolved by coupling our ATAS beamline[27,32] with an in-situ time-of-flight mass spectrometer (TOF-MS, Fig. 3C). This device identifies the ionic fragments (Fig. 3D) generated by the pump pulse independently of experimental and simulated absorption spectra (see Section S2.2). By tuning the pump pulse intensity through an in-vacuum motorized iris,

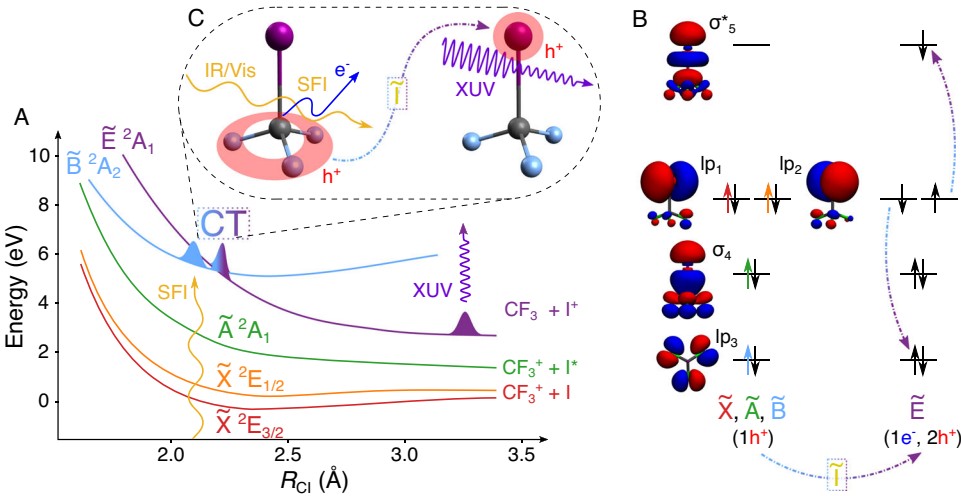

**Fig. 2 | CF₃I⁺: a molecular system manifesting a charge-transfer crossing. A** A schematic diabatic representation of the PECs of CF₃I⁺ showing the initialization of the dynamics through SFI at the FC point and the nearby crossing of the B̃ and Ẽ states. **B** The leading character of the electronic states is shown with an orbital energy diagram (the respective ionized electrons of each state are shown as colored arrows). The passage between the B̃ and Ẽ states requires the rearrangement of two electrons and therefore we postulate that it involves coupling to one or several intermediate states with ²E-symmetry (collectively labeled Ĩ). **C** This rearrangement moves hole density from the fluorine atoms to the iodine, which is observed through the site specificity of ATAS at the iodine N₄,₅ edge.

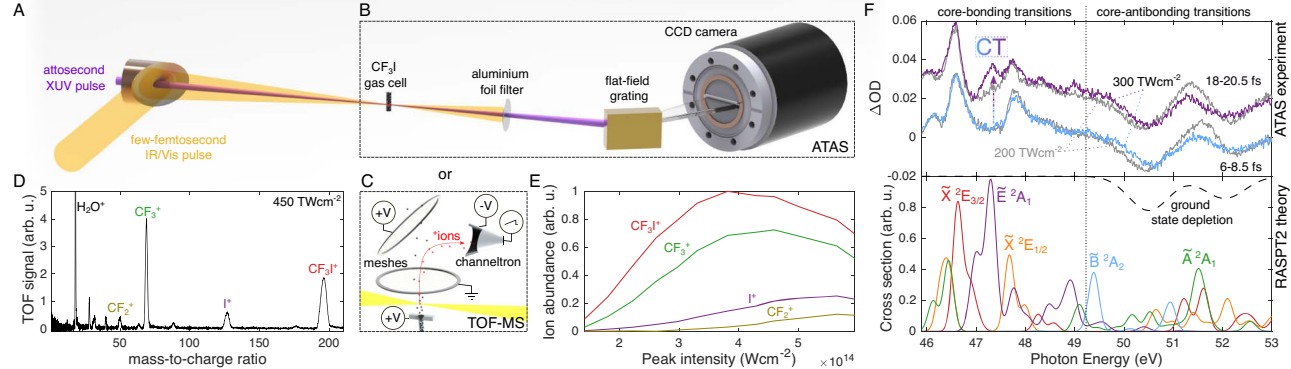

**Fig. 3 | Experimental methodology and overview of results. A** An attosecond beamline[32] delivers an isolated attosecond XUV pulse and a few-femtosecond near-infrared/visible (IR/Vis) pulse with a tunable delay. These are used to perform **B** ATAS[27], or **C** TOF-MS. **D** Mass spectrum measured at a peak intensity of 450 TW·cm⁻². The relative intensity-dependent abundances of the major fragments are shown in (**E**). **F** Two transient spectra from two ATAS experiments at different pump intensities (top) are compared with energy-shifted RASPT2 ab-initio spectra (see Section S3.4) computed at a C−I bond length of 2.34 Å (bottom). Note that the B̃ state is not populated at this bond length and should be absent in experimental spectra. An intensity- and delay-dependent feature is observed at 47.3 eV corresponding to the CT Ẽ state.

we acquire intensity-resolved ion yields and transient-absorption spectra (Fig. 3E, F), which allow us to determine the optimal peak intensity for investigating the B̃ state dynamics with ATAS. An intensity of 300 TW·cm⁻² gives a large fraction of I⁺ (and therefore the initial B̃ state population) while minimizing the abundance of fragments like CF₂⁺, which originate from higher-lying excited states.

The experimental results are interpreted with the help of state-of-the-art restricted-active-space self-consistent-field (RASSCF) methodology[33,34], used to calculate the valence-excited states of CF₃I⁺ as well as their core-valence absorption spectra, while also capturing the effects of SOC. The general protocol for calculating these states and transitions up to second-order perturbation theory (RASPT2) is discussed in ref. 35 and Section S3. The specific protocol applied for the study of CF₃I⁺ is discussed in the "Methods" section.

Figure 3F compares the transient-absorption spectra measured at two pump-probe delays and pump intensities with the RASPT2 calculations, demonstrating quantitative agreement. At the lower (200 TW·cm⁻²) pump intensity, where only the spin-orbit-split X̃ ²E ionic ground state is excited (gray line), the core-bonding region (<49 eV)

exhibits four absorption features; two weaker at 46.1 and 48.4 eV and two stronger at 46.6 and 47.8 eV. Their respective splittings correspond to the 1.7 eV (0.6 eV) intervals of the 4d (5p) levels of iodine. The absence of observable signatures of the Ã state is discussed in Section S4.

When the pump intensity is increased to 300 TW·cm⁻², a new delayed absorption feature appears at 47.3 eV (highlighted with "CT" in Fig. 3F) along with weaker features between 48 and 49.5 eV. These match the RASPT2 spectrum of the Ẽ state, as well as previous measurements of the N₄,₅ absorption of I⁺[36] (to which the Ẽ state dissociates). The agreement of experiment, literature, and calculations allows for the unambiguous assignment of the absorption peak at 47.3 eV to the Ẽ state.

Since the hole density in the B̃ state is localized on the fluorine atoms with very little spatial overlap with the iodine 4d orbitals, its spectrum contains minimal core-to-hole-type transitions, which dominate the N₄,₅ spectra of the X̃, Ã and Ẽ states. Instead, similar to the CF₃I neutral ground state (the spectrum of which can be found in Figs. S7 and S8), the B̃-state spectral features are weaker and limited to

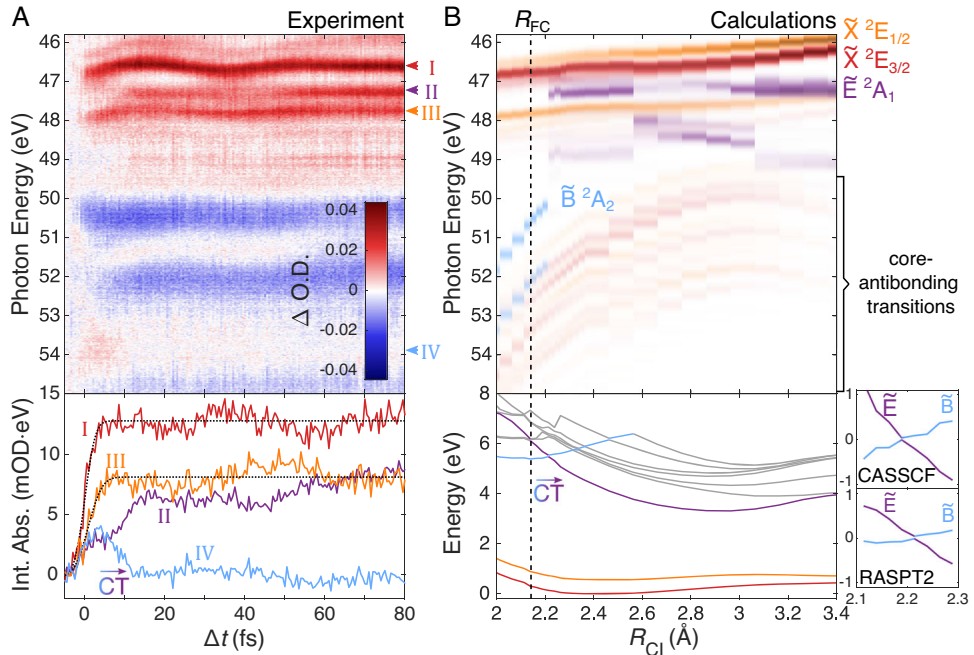

**Fig. 4 | Detailed analysis of ATAS results. A** The top panel shows the ATAS results of the high-intensity measurement (300 TW·cm$^{-2}$). Colored arrowheads on the right identify the spectral features whose integrated transient absorption is investigated in the bottom panel (see the main text for the spectral ranges used). **B** The top panel shows the energy-shifted RASPT2 $R_{CI}$-resolved absorption spectra (see Section S3.4) of the relevant cationic states. The spectra of the different states are differentiated by color and the FC point ($R_{FC}$ = 2.14 Å) is marked with a vertical dashed line. A quantitative comparison using a common colorscale is provided in Fig. S14. In the bottom panel, the adiabatic PECs of CF$_3$I$^+$ are shown, color-coded according to their leading electronic character (see Table S4). The $\tilde{B}$ / $\tilde{E}$ crossing at $R_{CI}$ = 2.21 Å is evident (note that the active space is not large enough to accurately describe the gray-colored states). The inset (right) shows this CT crossing in more detail for the CASSCF and RASPT2 calculations.

the core-antibonding transitions, which are found at spectral energies > 49.5 eV (as shown in Fig. 3F).

To identify these $\tilde{B}$-state absorption features and investigate their temporal evolution, we turn to Fig. 4, which compares the high-pump-intensity ATAS results (A) with the calculated absorption spectra of the first, second, and fourth adiabatic states (B). The calculated spectra are also plotted separately using a common color scale in Fig. S14. Time zero ($\Delta t$ = 0 fs) is set to the center of the rise of the 46.7-eV $\tilde{X}^2E_{3/2}$-state absorbance, determined from an error function fit $0.0 \pm 0.3$ fs (see bottom of Fig. 4A). The previously discussed onset of the $\tilde{E}$-state absorption around 10 fs is again evident. This is also reproduced in the RASPT2 calculations, where the fourth adiabat only exhibits $\tilde{E}$-state absorption after the CT crossing at $R_{CI}$ = 2.21 Å.

Figure 4A also reveals the presence of a weak, broad, and short-lived absorption feature between 53.5 and 55 eV, which appears absent in ATAS measurements at 200 TWcm$^{-2}$. Although slightly shifted compared to the calculated $\tilde{B}$-state spectra (which we attribute to the differences between the real CT reaction path and the geometries used for the calculations, as well as the $\tilde{B}$ state's unknown energy shift—see section S3.4) this feature disappears just as the absorption at 47.3 eV rises, indicating that population is being transferred between the two states. The bottom panel of Fig. 4A shows this anti-correlation more clearly, plotting the integrated transient absorption as a function of time for the regions 46.36–47.09 eV (I), 47.09–47.53 eV (II), 47.53–48.12 eV (III), and 53.50–54.30 eV (IV). This provides sufficient evidence to assign feature IV to the $\tilde{B}$ state.

To make the direct connection between the experimental absorption observable and diabatic populations (and eventually a delay), we must establish the invariance of the absorption cross section with respect to C–I coordinate in the vicinity of the CT crossing. This is confirmed in section S3.7 by spectrally integrating the RASPT2 results in and vicinity of the crossing, confirming that regions II and IV can be used as measures of the transient diabatic populations of the $\tilde{B}$ and $\tilde{E}$

states. The population transfer delay can now be experimentally investigated.

We extract the experimental delay in the same manner as for the results of the simulations; by fitting the CDF of the AGND to the transient absorption (details in Section S2.4). The resulting location parameters $\mu_{\tilde{B}} = 8.65 \pm 0.34$ fs and $\mu_{\tilde{E}} = 10.11 \pm 0.23$ fs reveal a significant population-transfer delay: $\tau = \mu_{\tilde{E}} - \mu_{\tilde{B}} = 1.46 \pm 0.41$ fs. While a two-state diabatic model is unable to replicate such dynamics, our three-state simulations confidently reproduce the delay (see Fig. 5B, C). The reaction coordinate $x$ maps onto the C–I bond length and the model parameters are adjusted to reflect the PECs of CF$_3$I$^+$. The initial vibrational wavepacket was set to the ground state of CF$_3$I (with the same width as the simulations in Fig. 1 but initially centered at −0.12 Å). The parameters of the Hamiltonian were $\alpha_0 = 0.25$, $\alpha = 0.2$, $\gamma = 0.045$ and $c = 0.04$. As in Fig. 1, the dynamics are almost entirely adiabatic, exhibiting a population transfer delay of 1.48 fs.

It is important to note that linear and quadratic vibronic coupling between the $\tilde{B}$, and $\tilde{E}$ states is symmetry-forbidden for geometries in the C$_{3v}$ point group because the direct product of the irreducible representations of these states with that of the C–I stretching mode are not totally symmetric. The CT crossing shown in the RASPT2 PECs is thus a true conical intersection. An adiabatic reaction can only proceed by passing around it, corresponding to symmetry breaking, which naturally occurs due to the finite extension of the vibrational wavepacket along non-totally-symmetric modes. Our three-state model can be viewed as describing the CT-reaction at such distorted geometries. A detailed symmetry analysis of the vibronic-coupling problem is provided in Section S5.

The experimental AGND fits allow us to also determine the vibrational rearrangement time of the CT reaction from the mean of the location parameters $(\mu_{\tilde{B}} + \mu_{\tilde{E}})/2 = 9.38 \pm 0.21$ fs. In doing so, we assume that the $\tilde{E}$ population is created at the same time as the cationic ground state. For different states of the same cation, this is normally

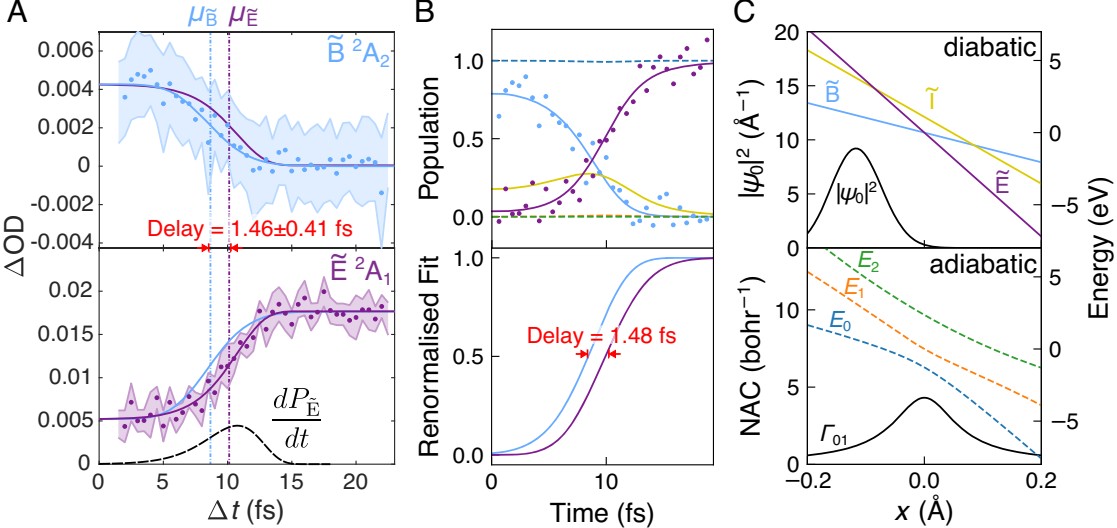

**Fig. 5 | Determination and theoretical reproduction of the charge-transfer delay. A** Transient absorption of the $\tilde{B}$ and $\tilde{E}$ states (dots, shaded error ranges represent the standard deviations) fit with the AGND CDF (solid lines). The location parameters, $\mu_i$, are shown as dashed-dotted vertical lines. Lines of opposite color to the dots show the inverted and rescaled fit of the other state to emphasize the delay. The AGND probability density function of $\tilde{E}$, corresponding to the rate of population transfer is plotted below as a dashed black line. **B** The top panel shows how the rescaled experimentally observed population dynamics (dots) can be reproduced with the three-state crossing model introduced in Fig. 1 (the diabatic (solid) and adiabatic (dashed) transient populations are shown). The delay in the population transfer is emphasized in the bottom panel, which shows the renormalized AGND fits of the simulated population dynamics. **C** The effective one-dimensional PECs (colored), initial probability distribution (black, top), and NAC (black, bottom) used for these calculations are shown.

correct to within a few-hundred attoseconds[37]. This is the delay at which the population of the intermediate state is maximized and the midpoint of the CT reaction. Meanwhile, the scale parameters $\sigma_{\tilde{B}} = 2.44 \pm 0.44$ fs and $\sigma_{\tilde{E}} = 2.34 \pm 0.36$ fs represent the times taken for the initial diabatic state to lose its population and for the final state to gain it, respectively. To date, such electronic population transfer timescales could not be deconvolved from the vibrational dynamics (rearrangement time) and only be inferred indirectly, e.g., through core-hole clock spectroscopy[19].

Starting with a general theoretical model we have shown the existence and elucidated the origin of a novel transient observable in state crossing dynamics: a delay in population transfer caused by the coupling of two crossing states through a third. Using attosecond spectroscopy supported by advanced quantum-chemistry calculations, we have also demonstrated its existence experimentally. Investigations of the theoretical model's parameters have revealed that such delays can be expected to be on the order of a single femtosecond for intersecting valence states of molecules, opening an exciting frontier for attochemistry.

The site specificity and sub-femtosecond temporal resolution achieved in this work have moreover enabled the direct real-time observation of the primary quantum-mechanical processes underlying molecular CT. Our measurements have determined the vibrational rearrangement time taken by the nuclear wave packet to reach the CT crossing ($9.38 \pm 0.21$ fs), the population-transfer times (2.3–2.4 fs), and a delay ($1.46 \pm 0.41$ fs) in population transfer. In contrast to charge migration, the timescale of which is simply determined by the energetic separations of the populated quantum states, the delay discovered in this work encodes diabatic couplings to additional states beyond the two intersecting states. These couplings are neither accessible to frequency-domain spectroscopy, nor straightforward to calculate. As such, these delays reflect non-trivial molecular properties and affect one of the most fundamental quantum processes in matter.

Looking forward, our work defines a set of methodologies for unraveling the primary quantum-mechanical processes that underlie all population-transfer dynamics, in particular long-range CT[38], which is known to play a fundamental role in photovoltaics[39],

photosynthesis[40,41], and photocatalysis[42]. The sensitivity of XUV/X-ray spectroscopy to electronic coherences[27,43] and its applicability to complex liquid-phase systems[44,45] make the present methodology relevant for probing and exploiting coherences in complex molecular systems featuring scientifically and technologically relevant charge-transfer dynamics, such as those underlying photosynthesis, photovoltaics, or molecular optoelectronics.

## Methods
### Theoretical methods
To solve the time-dependent Schrödinger equation (TDSE) for the 3-state model system presented in our work, we use the split-operator method. Convergence of the simulation results was verified with respect to grid size and time step. A grid spacing of 0.002 a.u. and a time step of 5 a.u. were found to be suitable. To ensure that the populations were conserved, no absorbing boundary conditions were employed, and the grid was made large enough that the wave packet did not reach the edge within the simulation time. For more details on the theoretical model, please see Section S1.

### Experimental methods
The ATAS experiment is driven using a 5.2-fs full-width at half-maximum optical pulse spanning the 500–1000 nm spectral range. This pulse is generated by spectrally broadening the compressed output of a FEMTOPOWER V CEP titanium:sapphire laser system. Serving dual purposes, the few-cycle pulse acts both as the pump and as the driving field for high harmonic generation (HHG). The HHG medium is argon, which produces attosecond XUV probe pulses characterized by a continuous spectrum at the iodine $N_{4,5}$-edge. A custom-built attosecond interferometer controls the relative delay between the pump and probe pulses.

Both pulses are directed through a gas target, which takes the form of a 3 mm tube, fed with an Even-Lavie pulsed valve. Upon passing the transient absorption target, the pump and probe pulses are separated with another aluminum-foil filter. The transmitted XUV is spectrally dispersed using a Hitachi 001-0660 *3 aberration-corrected concave grating (positioned with the TAS target at its imaging focus)

and detected using a Princeton Instruments PIXIS-XO:2KB backlit-CCD, providing a resolving power of over 1000.

The same few-cycle pump pulse employed in the ATAS experiment is also utilized for in-situ mass spectrometry. For this purpose, the target chamber is filled with $CF_3I$ via the ATAS cell, and a high voltage is applied to induce acceleration of cations produced by strong-field ionization. These ions are subsequently detected by a Photonis MigaSpiraltron channeltron, which records their time of arrival. For more details on the experimental procedure and the data analysis, please see Section S2.

## Computational methods

To calculate the valence- and core-excited states of $CF_3I^+$, a restricted active space RAS (34, 1, 1; 5, 13, 3) was utilized. The orbital space was divided into three sub-spaces, RAS1, RAS2, and RAS3. As the ATAS probed the $N_{4,5}$ edge of iodine, its five $4d$ orbitals were included in the RAS1, and a maximum of one electron was allowed to be excited out of this sub-space. The RAS2 was comprised of the carbon-iodine bond ($\sigma_4$, $\sigma_5^*$), both iodine lone-pair orbitals $lp_1$ and $lp_2$ as well as the six fluorine lone-pair orbitals $lp_3$, $lp_4$, $lp_5$, $lp_6$, $lp_7$, and $lp_8$. Additionally, it included the three carbon-fluorine bonds $\sigma_1$, $\sigma_2$, and $\sigma_3$, but to reduce computational costs, their corresponding virtual orbitals ($\sigma_6^*$, $\sigma_7^*$ and $\sigma_8^*$) were included in the RAS3 sub-space and only a single excitation into them was allowed, while for the orbitals in the RAS2 all possible configurations were taken into account. The sizes, as well as the composition, of the three sub spaces was thoroughly tested and the details of this benchmark are presented in Section S3.2.

To obtain the final ab-initio spectra, two different RASSCF calculations (one for the valence-excited and one for the core-excited states) were performed. The core-excited states were calculated utilizing core-valence-separation. Both calculations included dynamic electron correlation perturbatively, through a multi-state complete active space perturbation theory (MS-CASPT2) calculation. As these types of calculations, especially for the core excited states, are prone to intruder states, we applied a ionization-potential electron-affinity and imaginary shift of 0.25 hartree and 0.2 hartree, respectively. Relativistic effects were included and treated in two steps, both based on the Douglas–Kroll Hamiltonian. For the first step all calculations were performed with the relativistic atomic natural orbital basis, contracted to ADZP quality (ANO-RCC-ADZP). In a second step, to include the SOC, all calculations of the valence and core excited states have been performed for both doublet and quartet multiplicities. These four wave functions were then used as the basis in a restricted active space state interaction (RASSI) calculation, where the interactions between all states are treated, including the SOC, by means of the atomic mean-field spin-orbit integrals. Finally, the excitation energies $\Delta E$ as well as the corresponding transition dipole moments of the core-valence transitions were extracted from the results of the RASSI calculation, and used to construct the absorption spectra. Further details on the protocol are provided in Section S3.3.

All calculations are performed along the optimized C–I stretching mode of the ionic ground state calculated at the SA11-CASSCF(17,13)/ANO-RCC-VDZP level of theory. Further details on the calculation of the PECs are provided in Section S3.4.

## Reporting summary

Further information on research design is available in the Nature Portfolio Reporting Summary linked to this article.

## Data availability

Source data are provided with this paper.

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

## Acknowledgements
We thank M. Belozertsev for his early contributions to the theoretical research and A. Schneider, M. Seiler and M. Urban for their technical support. D.T.M., P. Z. and H.J.W. gratefully acknowledge funding from ETH Zürich. F.R., T.S. and R.d.V. acknowledge funding from the DFG Normalverfahren and the computational and data resources provided by the Leibniz Supercomputing Centre (www.lrz.de). Z.L. acknowledges funding from National Natural Science Foundation of China (project no. 12174009, 12234002).

## Author contributions
H.J.W. and D.T.M. conceived the experimental study. D.T.M. and P.Z. performed the experiments and D.T.M. analyzed the results. D.T.M. and J.O.R. performed the three-state model simulations. F.R and T.S. performed, analyzed and post-processed the RASPT2 and DFTMRCI calculations. H.J.W. supervised the experimental work. T.S. and R.d.V. supervised and coordinated the quantum-chemistry part of this work. Z.L. and D.T.M. performed preliminary quantum-chemical calculations. D.T.M., F.R., T.S., J.O.R., H.J.W. and R.d.V. discussed the results, finalized the interpretations and wrote the paper. All authors reviewed and/or edited the paper.

## Competing interests
The authors declare no competing interests.
