## [Transparent Peer Review file · Nature Communications]

Attosecond spectroscopy of molecular charge transfer uncovers a 1.5-fs delay in population transfer

Corresponding Author: Professor Hans Jakob Wörner

Version 0:

Reviewer comments:

Reviewer #1

(Remarks to the Author)

This paper deals with a very old problem in Molecular Physics, i.e. the transfer of electronic population between two states in the presence of a third one. To quote an older example, in the 90s there was an intense debate of the question whether the initial electronic transfer in bacterial photosynthesis proceeds step-wise or via superexchange. Using traditional valence state spectroscopy ultimate answers are difficult since details are buried underneath broad lineshapes and kinetics is extracted via spectral decomposition. With this being said it is clear that deeper insight requires not only sufficient time resolution but also spectral selectivity. This is what attosecond Vis-XUV spectroscopy offers as is shown in this manuscript for a simple model reaction.

To make it short this is an excellent paper which reports real ground breaking results. I cannot judge on tiny details of the experimental setup, but overall the steps taken for validation appear to me rather plausible. In addition high level theory is used to confirm conclusions (assignments). Therefore I recommend publication.

A few minor points:

- κ and μ should briefly be introduced in the main text

- The sentence "The calculations are performed along the optimized C–I stretching mode of the ionic ground state which, importantly, is not equivalent to the \tilde{B}/\tilde{E} CT reaction coordinate." on p.8 is a bit disturbing. A brief comment on the relevance of this distinction in the present context would help.

Reviewer #2

(Remarks to the Author)

This manuscript reports the use of attosecond XUV transient absorption spectroscopy to observe a 1.46-fs delay between the depopulation of the B state and the population of the E state in the CF₃I⁺ ion. The experimental result is supported by theoretical simulations that solve the time-dependent Schrödinger equation for either two or three diabatic electronic states as well as RASPT2 ab initio calculations that yield the potential energy curves and XUV absorption spectra of the various species involved in the C–I dissociation process. The reported results are intriguing, and if valid, could be of interest to the Nature Communications readership. Several major issues concerning the interpretation of the results should be addressed, however, before a recommendation on publication can be made.

1. It is known from sequential chemical kinetics that population from state A flowing through state B before reaching state C will result in a delayed appearance of population in state C. It is not clear how the results reported in the manuscript differ from this standard piece of textbook knowledge. Moreover, the authors invoke the existence of an intermediate state, I, in the theoretical calculations and in the interpretation of the experimental results. However, the identity of state I is not provided.

2. A relatively simpler explanation for the observed time delay between the depletion of the B state signal and the rise of the E state signal is that the XUV transition energies that are used in the analysis probe slightly different nuclear configurations,

i.e., they have different probe windows. It seems that the authors have not considered this possibility.

3. Much of the interpretation of the experimental results relies on analyzing a weak signal that appears in the vicinity of pump-probe overlap at ~54 eV. To complicate matters, the authors had to apply SVD to clean up a rather noisy dataset (Fig. S4). The use of SVD calls into question if the spectral feature at ~54 eV could be an artifact of data processing. The authors should furnish more information regarding the SVD filtering, e.g., the first 20 singular values and the delay vectors for indices 11, 12, 13, and 14 for each of the three delay scans, to justify the choice of SVD parameters used.

4. In the vicinity of pump-probe overlap, additional laser-molecule interactions such as ac Stark shift can occur. This is especially so when high peak intensities of 1014 W/cm² are used in the experiments. Have the authors considered the possibility that such nonlinear interactions could give rise to the weak signal at ~54 eV? A useful piece of data to show would be the ATAS data, similar to that shown in Fig. 4A, but taken at lower peak intensities, e.g., 200 TW/cm², at which the authors claim that only the electronic ground state of the ion is populated.

5. The authors uncover a time delay of 1.46 +/- 0.41 fs from their analysis. This time delay, however, is probably dependent on the mathematical function that is used to fit their data. Can the authors comment on the degree of sensitivity of the time delay to the mathematical function that is used to fit the experimental data? In addition, it would also be useful if the authors could show why they do not simply use exponential functions to fit their data, beyond arguing that their high time resolution permits them to go beyond simple kinetic models. (Besides, it is not clear that the use of the AGND function is physically motivated, beyond obtaining an time-asymmetric response.) Related to this point, it would be interesting to know what time delay is obtained when simple exponential functions are used to fit the data.

6. The authors report a vibrational rearrangement time of 9.38 +/- 0.21 fs that accompanies the charge transfer reaction. The readers would benefit from elaboration on the relation between the mean of the location parameters and the vibrational rearrangement time. Moreover, the time is referenced to the midpoint of the rise of the ground state ion signal. Presumably formation of the B state requires higher intensities and hence occurs at a different time along the strong-field-ionizing laser pulse envelope. In this case, what is the significance of the quoted vibrational rearrangement time?

7. The authors rightly point out that the B state has little iodine character and therefore can only be detected via transitions from the I 4d level to Rydberg states. These transitions appear in the >50 eV region. In this region, one should also find core-to-Rydberg transitions for the other states of CF₃I⁺. However, it is unclear if these transitions are observed. Furthermore, strong-field ionization should also yield the A state, which is expected to undergo rapid C-I dissociation according to the potential energy curve shown in Fig. 2A. However, it seems that neither the A state nor its I* dissociation product is detected in the experiment.

8. The authors claim that their assignment of the 47.3 eV peak to the E state of CF₃I⁺ is "unambiguous". As the authors point out, this transition energy agrees with the I+ 4d absorption at 47.32 eV (ref. 33 in the manuscript), but why does that support its assignment to the E state? Why should the XUV transition energies of the E state and the I+ ion be identical? Moreover, bond dissociation along the C-I coordinate, first for the B state and subsequently for the E state, should lead to a time-dependent shift in the XUV transition energies of these states, which is not observed; instead, the transition energies of the absorption at ~54 eV and 47.3 eV remain largely unchanged as a function of time delay.

Reviewer #3

(Remarks to the Author)
Review attached.

Reviewer #4

(Remarks to the Author)
I co-reviewed this manuscript with one of the reviewers who provided the listed reports. This is part of the Nature Communications initiative to facilitate training in peer review and to provide appropriate recognition for Early Career Researchers who co-review manuscripts.

Version 1:

Reviewer comments:

Reviewer #1

(Remarks to the Author)
I considered this paper as excellent work already in my previous review. My minor points have been properly addressed. In addition I had a look at the comments of the other reviewers and the related response by the authors. In my view, the authors properly addressed the points and clarified some misunderstandings. Of course, there could be room for improvement of the theoretical model, but this would go beyond what's needed to support the main conclusions. I recommend publication.

Reviewer #2

(Remarks to the Author)

I greatly appreciate the authors' detailed replies to the reviewers' comments. In most cases, the replies are satisfactory. However, the identity of the intermediate state remains unclear. While they have undertaken a symmetry analysis to show that the intermediate state has 2E symmetry, they have yet to identify the state. Maybe another way to phrase my question is to ask which of the potential curves in the lower panel of Fig. 4B is that of the intermediate state, or which of the 2E states in Table S2 is the intermediate state?

Reviewer #3

(Remarks to the Author)

We thank the authors for their answers to the reviewers and for their clarifications, especially in the response letter. Indeed, we still believe that theoretical validation of the existence of the state(s) I based on quantum chemistry would make the observations reported in this work more robust (and it does not seem at all beyond the scope of the present work). However, we appreciate the originality of the experiment and of its analysis reported in the manuscript and we believe that it has indeed the potential to stimulate further studies. Therefore, we can recommend the manuscript for publication in Nature Communications.

Reviewer #4

(Remarks to the Author)

Version 2:

Reviewer comments:

Reviewer #2

(Remarks to the Author)

I am glad to hear that the authors have successfully converged the calculations and identified the intermediate state. This is a very nice piece of work. I recommend publication of the manuscript in its current form.

Reviewer #3

(Remarks to the Author)

We thank the authors for the thorough analysis of the electronic structure of the system and we very much appreciate their efforts to validate their experimental observations. We find the work suitable for publication in its current form.

Reviewer #4

(Remarks to the Author)

The manuscript *Attosecond spectroscopy of charge transfer uncovers a 1.5-fs Delay in Population Transfer* by Wörner and co-workers reports a combined theory-experiment investigation of the time delay in the electronic population transfer during the dissociation dynamics of CF_3I^+ initiated by photoionization. The dynamics is probed via attosecond transient absorption spectroscopy, allowing for a sub-femtosecond resolution and, thus, it is possible to resolve a population transfer delay of the order of 1.5 fs.

In the first part of the manuscript, the concept of population transfer delay is introduced using a quantum-dynamical picture on a one-dimensional model. The model consists of 3 electronic diabatic states $|0\rangle$, $|1\rangle$, $|2\rangle$ and is used to represent population transfer from $|0\rangle$ to $|2\rangle$. However, the states $|0\rangle$ and $|2\rangle$ are not diabatically coupled, but $|0\rangle$ is coupled to $|1\rangle$ and $|2\rangle$ is coupled to $|1\rangle$, such that population transfers from $|0\rangle$ to $|2\rangle$ via the “intruder” state $|1\rangle$. Such transfer is delayed with respect to a direct transfer from $|0\rangle$ to $|2\rangle$. Such a delay is estimated of the order of 1 to 2 fs using typical molecular valence state parameters in the model. In the second part of the manuscript, the electronic structure of CF_3I^+ is analyzed using the ab-initio multiconfigurational second-order perturbation theory restricted active space (RASPT2) method, and the experimental signals, namely the transient absorption spectra, are interpreted based on the RASPT2 calculations. In the third part of the manuscript, the transient absorption signal is interpreted based on the model introduced in the first part.

While we believe that experiments with such a temporal resolution are able to highlight *some* time delay during the charge transfer process occurring upon photoionization, we are not convinced that it can be interpreted as done by the authors, namely by hypothesizing the presence of an “intruder” electronic state which neither is observed in experiments nor appears in the calculations. Therefore, at this stage, we cannot recommend publication of the manuscript in Nature Communications and we invite the authors to address the following points.

1. The interpretation of the time delay in the population transfer between \tilde{B} and \tilde{E} in terms of the “intrusion” of an additional state \tilde{I} that cannot be observed, by the quantum chemistry calculations and by experiments, is confusing and perhaps not even necessary. The analysis of the occupation of the molecular orbitals and the single electron picture used to describe the change of electronic character (Fig.2) suggest a two-electron rearrangement process that is certainly not instantaneous. Can it be that the authors observe this delay in their experiment?
2. As follows from point 1., the dynamical picture used to introduce the concept of population transfer delay and to validate the experimental observations is not convincing. If the authors strongly believe that this interpretation is, instead, robust, we invite them to find the state(s) \tilde{I} at least with quantum chemistry calculations.
3. While the analysis of the couplings between \tilde{B} , \tilde{E} and \tilde{I} based on symmetry considerations holds valid at the C_{3v} equilibrium geometry along the elongation of the C-I bond, the ab initio dynamical picture of the molecule is quite different, and symmetry is probably broken during the dynamics. Therefore, distorted geometries should be investigated.
4. The results presented in Fig.4 are not convincing. First of all, in the upper panel of B the authors do not explain how the observed transient absorptions (reported in A) are “rescaled” to fit into the diabatic populations dynamical picture – which is the picture used to introduce the concept of population transfer delay (Fig.1) in the manuscript. While in Fig.1 the presence of non-zero population P_1 at time 0 is acceptable since it is just a theoretical model, in panel B of Fig.4 the authors need to explain how the state \tilde{I} is populated by the excitation. It seems to us that the dynamics is initialized by fully populating the adiabatic ground state (dashed blue curve in C bottom

panel), which then has contributions in the 3 diabatic states. However, this is somehow counterintuitive when comparing theory with experiments because the bright electronic states in the Franck-Condon region are usually identified based on symmetry arguments on the diabatic states.

5. The population transfer delay is referred to as an *observable* in the manuscript. However, it can be deduced only in the diabatic picture while it seems to disappear in the adiabatic picture, as the population indicated in dashed blue in Fig.4B upper panel is always equal to 1 since the process is fully adiabatic. How do the authors justify the fact that this *observable* depends on the representation?

Minor points:

6. An explanation of the reason why in Fig.3F (lower panel) the C-I bond length of 2.34 Angstrom is chosen to compute the spectrum should be provided.
7. Can the authors justify more clearly how the transient absorption of states \tilde{B} and \tilde{E} reported in Fig.4A gives direct access to the populations of these states? The fact that their initial values are basically the same seems to not support the interpretation as one state being not populated at the beginning.

Response to the referees

We would like to thank the referees for their valuable time they have taken to review our manuscript and their insightful comments and for their excellent suggestions, which helped us to further improve our manuscript. Below, we have color coded each referee's comment in blue, our replies in black and additions/changes in red in our resubmission. We hope that the referees will find our answers and the revised manuscript satisfactory.

Reviewer #1 (Remarks to the Author):

This paper deals with a very old problem in Molecular Physics, i.e. the transfer of electronic population between two states in the presence of a third one. To quote an older example, in the 90s there was an intense debate of the question whether the initial electronic transfer in bacterial photosynthesis proceeds step-wise or via superexchange. Using traditional valence state spectroscopy ultimate answers are difficult since details are of buried underneath broad lineshapes and kinetics is extracted via spectral decomposition. With this being said it is clear that deeper insight requires not only sufficient time resolution but also spectral selectivity. This is what attosecond Vis-XUV spectroscopy offers as is shown in this manuscript for a simple model reaction.

To make it short this is an excellent paper which reports real ground breaking results. I cannot judge on tiny details of the experimental setup, but overall the steps taken for validation appear to me rather plausible. In addition high level theory is used to confirm conclusions (assignments). Therefore I recommend publication.

A few minor points:

- kappa and mu should briefly be introduced in the main text

We have included a brief description of the parameters by replacing the following sentences on page 4

“To investigate these systematically, they are fit with cumulative distribution functions (CDFs) of asymmetric generalized normal distributions (AGNDs)~\cite{Hosking1997}. These functions are chosen to capture the temporally-asymmetric nature of the rate of population change, which LZSM-like transitions are well known for.”

with

“ To investigate these systematically, they are fit with cumulative distribution functions (CDFs) of asymmetric generalized normal distributions (AGNDs)~\cite{Hosking1997} whose parameters are the location μ , the scale σ , and the shape κ . μ specifies the median of the distribution and κ the degree of skewness. As a result, these functions can capture the temporally-asymmetric nature of the rate of population change, which LZSM-like transitions are well known for.”

- The sentence “The calculations are performed along the optimized C–I stretching mode of the ionic ground state which, importantly, is not equivalent to the \tilde{B}/\tilde{E} CT reaction coordinate.” on p.8 is a bit disturbing. A brief comment on the relevance of this distinction in the present context would help.

Thanks to the large spin-orbit coupling of iodine (which is larger than the Jahn-Teller stabilization energy in the present case), the X ground states of the CF₃I cation has a stable equilibrium geometry of C_{3v} symmetry. We find that the optimised C-I stretching mode of the X_{3/2} ground state also preserves this C_{3v} symmetry. On the other hand, we find in section S5 that symmetry breaking through asymmetric distortions of the CF₃ moiety is required for the B, E and I states to couple and the CT reaction to occur. This means that the CT reaction actually takes place at slightly distorted geometries, rather than exact C_{3v} geometries. Since there is no mechanism that breaks the overall symmetry of the wave packet, the center of mass of the wavepacket actually follows the C_{3v}-symmetric minimum-energy pathway. This situation is reminiscent of all conical-intersection dynamics where population transfer does not take place for exclusive motion along the tuning mode, but necessitates a displacement along the coupling mode.

To avoid confusion, we have removed the phrase “which, importantly, is not equivalent to the \tilde{B}/\tilde{E} CT reaction coordinate” and we have added “Further details on the calculation of the PECs and symmetry considerations are provided in Sections S3 and S5, respectively.”, where the detailed arguments are provided.

Reviewer #2 (Remarks to the Author):

This manuscript reports the use of attosecond XUV transient absorption spectroscopy to observe a 1.46-fs delay between the depopulation of the B state and the population of the E state in the CF3I+ ion. The experimental result is supported by theoretical simulations that solve the time-dependent Schrödinger equation for either two or three diabatic electronic states as well as RASPT2 ab initio calculations that yield the potential energy curves and XUV absorption spectra of the various species involved in the C–I dissociation process. The reported results are intriguing, and if valid, could be of interest to the Nature Communications readership. Several major issues concerning the interpretation of the results should be addressed, however, before a recommendation on publication can be made.

1. It is known from sequential chemical kinetics that population from state A flowing through state B before reaching state C will result in a delayed appearance of population in state C. It is not clear how the results reported in the manuscript differ from this standard piece of textbook knowledge. Moreover, the authors invoke the existence of an intermediate state, I, in the theoretical calculations and in the interpretation of the experimental results. However, the identity of state I is not provided.

The referee is correct that population starting in state A, flowing through state B, into state C can be expected to exhibit a delay when observed in the time domain. This is the case for any process which can be described in terms of time-dependent populations and therefore even more general than chemical kinetics. However, ultrafast dynamical processes can often not be described by rate processes because the assumptions made for the derivation of kinetic rate laws are usually not fulfilled.

Our results are novel in two ways. Firstly, such a delay has never been observed nor discussed for electronic processes. Experimentally resolving the electronic character during the state crossing on a sub-femtosecond timescale has allowed this delay to be observed and is the first experimental methodology that can detect the participation of a third state in an electronic state crossing. Secondly, the dynamics are very clearly non-exponential, which prevents the chemical-kinetics textbook knowledge from being applied here, a point that is raised in the other comments and that we will return to below. We politely disagree with the statement that we do not identify the intermediate state. Thanks to the sophisticated quantum chemistry calculations, which clearly identify the configuration of the initial and final states, we are able to identify the configuration and symmetry of the intermediate state. A detailed discussion of this can be found in section S5.

2. A relatively simpler explanation for the observed time delay between the depletion of the B state signal and the rise of the E state signal is that the XUV transition energies that are used in the analysis probe slightly different nuclear configurations, i.e., they have different probe windows. It seems that the authors have not considered this possibility.

The referee is correct insofar as regions IV and II (as labelled in Fig. 4) probe different nuclear configurations: the first probes the population before the CT crossing and the second the population after the CT crossing. This, however, does

not explain why a delay is observed between them: a direct interaction between the B and E state (which is symmetry forbidden at C_{3v}-symmetric geometries) would not exhibit a delay, implying that a third state must be involved (see Section S5 for the detailed symmetry arguments).

If the referee's point is that the delay appears because the absorption of the excited state population leaves the chosen regions of integration, they are also correct: the total population must be conserved, meaning that the absorption of the intermediate state must appear somewhere in the spectrum (provided that the corresponding excitation is allowed). This, however, only rephrases the question and does not explain the reason for the delay / "temporary loss of absorption from the expected regions of interest".

If the referee's comment is meant to express that the delay is due to purely vibrational dynamics, we are happy to reassure them that we have confidently eliminated this mechanism. This has been excluded based on both experimental and quantum chemical results. Experimentally, we see that there is no continuous change of the absorption between the B and E states: the absorption of region III remains unchanged during the CT process. Quantum mechanical results support this fact, showing a discontinuous evolution of the transition energies but an unchanged total absorption strength in both spectral regions of interest (see Section S 3.6 and Fig. S10). We have thus indeed considered the possibility mentioned the by reviewer and have excluded it.

To clarify this aspect, we have added the following sentence to the manuscript (p. 10-11): "To make the direct connection between the experimental absorption observable and diabatic populations (and eventually a delay), we must establish the invariance of the absorption cross section with respect to C–I coordinate in the vicinity of the CT crossing. This is confirmed in section S3.6 by spectrally integrating the RASPT2 results in and vicinity of the crossing, confirming that regions II and IV can be used as measures of the transient diabatic populations of the \tilde{B} and \tilde{E} states. The population transfer delay can now be experimentally investigated."

3. Much of the interpretation of the experimental results relies on analyzing a weak signal that appears in the vicinity of pump-probe overlap at ~54 eV. To complicate matters, the authors had to apply SVD to clean up a rather noisy dataset (Fig. S4). The use of SVD calls into question if the spectral feature at ~54 eV could be an artifact of data processing. The authors should furnish more information regarding the SVD filtering, e.g., the first 20 singular values and the delay vectors for indices 11, 12, 13, and 14 for each of the three delay scans, to justify the choice of SVD parameters used.

While the magnitude of the transient absorption signal of the B state is not large, its broad nature allows its integral, which is the quantity of interest, to be confidently evaluated. The feature is also present well beyond pump-probe overlap, which Fig. S2 shows is negligible for delays above 3 fs.

Regarding the referees concerns with the singular value filtering applied to the data, we would like to reassure them that this approach has been well tested and demonstrated in previous published work (see references in section S2.3). It is analogous to other correlation-based filtering methods that are commonly used in ATAS [<https://doi.org/10.1364/OE.435008>] and necessary to suppress the

well-understood fluctuations of the XUV spectrum that otherwise obfuscates time-dependent spectral changes.

Like any noise-suppression algorithm, singular-value filtration can introduce artifacts when used incorrectly. Nevertheless, we would like to emphasise that because it functions by removing only a few (normally 4-8) singular vectors, it is very robust against the introduction of the kind of time-frequency artifacts that would yield an artificial population transfer delay.

The figure on the next page shows the first 12 temporal and spectral vectors, which are the only vectors that are considered for removal by the algorithm. The other vectors are all retained and are therefore present in the results shown in the paper. The figure shows the temporal vector of each scan (dotted lines) as well as their mean (blue). The “reproducibility” metric (here called the “reliability”) evaluated from these temporal vectors is also shown. As vectors 1, 2, 3, 4, 6, 10 and 11 all have a reproducibility that falls below the chosen value of 0.1, they are removed from the dataset. By examining their corresponding spectral vector, we see that these all represent undesired spectrally-broad features which describe the random fluctuations in the spectrum of the broad XUV super-continuum and not the (spectrally much sharper) dynamics of the system.

We have also added below a plot (on the next page) showing the reproducibility as a function of the SVD index. This is our primary tool for determining the values of the two filtration parameters, i.e. the reproducibility cutoff (dashed red line) and the cutoff index (here set to 12, although the effect of the algorithm would not be changed if this was set anywhere in the range of 12-20 for this dataset). The separation of the vectors containing the information of interest, and those dominated by the random laser fluctuations are evident.

4. In the vicinity of pump-probe overlap, additional laser-molecule interactions such as ac Stark shift can occur. This is especially so when high peak intensities of 10^{14} W/cm² are used in the experiments. Have the authors considered the possibility that such nonlinear interactions could give rise to the weak signal at ~ 54 eV? A useful piece of data to show would be the ATAS data, similar to that shown in Fig. 4A, but taken at lower peak intensities, e.g., 200 TW/cm², at which the authors claim that only the electronic ground state of the ion is populated.

The effects of the pump field have been extensively considered. The degree to which the Stark shift influences the states of interest in this study is best inferred from the Xenon ATAS results in Fig. S2.; since Xenon and Iodine are of similar size, electronic structure and polarisability, their cationic states can be expected to exhibit a similar AC Stark shift. The way in which this influences the XUV absorption spectrum is discussed very nicely in <https://iopscience.iop.org/article/10.1088/1361-6455/ac9872>: a weak blue-shift shift of the absorption features is observed along with an instantaneous broadening of the signal which beats at twice the laser frequency and is absent for delays over 3 fs due to the decay of the pump field. Neither of these effects can account for a lasting absorption at 54 eV that shows no signs of 2-omega beats and is present up to a delay of 10 fs and appears at the same delay as the absorption features of the ground state. We can therefore safely exclude AC Stark shifts as the explanation for the weak signal at 54 eV.

5. The authors uncover a time delay of 1.46 ± 0.41 fs from their analysis. This time delay, however, is probably dependent on the mathematical function that is used to fit their data. Can the authors comment on the degree of sensitivity of the time delay to the mathematical function that is used to fit the experimental data? In addition, it would also be useful if the authors could show why they do not simply use exponential functions to fit their data, beyond arguing that their high time resolution permits them to go beyond simple kinetic models. (Besides, it is not clear that the use of the AGND function is physically motivated, beyond obtaining an time-asymmetric response.) Related to this point, it would be interesting to know what time delay is obtained when simple exponential functions are used to fit the data.

We first address the comments regarding the use of exponentials to fit the data and the treatment of the problem with chemical kinetics. Chemical kinetics employs rate equations and finds that the population dynamics of distinct chemical species in a reaction can be modelled with a combination of exponential functions. In doing so, the chemical reactions that inter-convert the different species are treated as semi-classical stochastic processes with time constants that mainly depend on the species' vibrational temperature and the energy barrier of the transition state (although dynamic information about the transition state or tunneling effects can be included in some theories). It is also important to note that while the kinetic treatment of reactions can be used to determine a difference in reaction rates, it is not possible to extract (or even define) a delay.

It is evident that the approximations of chemical kinetics are not valid for the ultrafast state crossing that we observe in our experiment: the different states are not distinct chemical species (they have a very similar nuclear arrangement that changes on much longer timescales than we consider in this work), there is no well-defined transition state, the state crossing occurs at a lower energy than the Frank Condon point, and it is impossible to define a temperature for a system so far from (local) equilibrium.

In agreement with these theoretical arguments, our experimental observables do not exhibit any exponential evolution; they are clearly sigmoidal functions which cannot be fit with exponential rises or decays.

We now discuss the functions that can capture the dynamics correctly. Ultrafast processes that occur ballistically or coherently exhibit two types of signals: oscillatory or sigmoidal. The former is treated with time-frequency analysis while the latter is typically fit with normal distribution functions, i.e. Gaussians and error functions. An example of such a fit can be found in Fig. S2 or in the work on Xenon cited in the next comment. The normal distribution has become the default fit for sigmoidal signals because their measurement has historically been limited by the cross-correlation of the ultrafast pulses employed, which is typically well-described by a Gaussian.

In the case of our experiment, however, the experimental time resolution ($\sigma = 1.0$ fs) is significantly shorter than the timescales of the dynamics we observe ($\sigma = 2.3/2.4$ fs). This allows us to go beyond the normal distribution and apply functions that might better fit the underlying dynamics. In this case, as we explain in more detail in Section S1.1, we determine this to be the AGND.

To summarise, the LZSM crossing (which can be solved analytically) informs us that the population transfer in a diabatic crossing should be significantly skewed (also supported by our 3-state TDSE simulations). To capture this and improve the robustness of our fits, we use the asymmetric extension of the normal distribution: the AGND.

For consistency, we apply the same AGND fit function to both the theoretical and experimental results. We can, however, replace the AGND fit with the normal distribution (equivalent to setting the skew parameter κ to zero). This normal fit yields a very similar result to the AGND: the position parameter for the B state is found to be 8.68 ± 0.40 fs and 10.22 ± 0.26 fs for the E state. The population transfer delay extracted from the normal distribution is then 1.54 ± 0.48 fs, very similar to the result from the AGND but exhibiting larger error bars because of a poorer fit. Our conclusions are therefore robust to the mathematical function employed.

6. The authors report a vibrational rearrangement time of 9.38 ± 0.21 fs that accompanies the charge transfer reaction. The readers would benefit from elaboration on the relation between the mean of the location parameters and the vibrational rearrangement time. Moreover, the time is referenced to the midpoint of the rise of the ground state ion signal. Presumably formation of the B state requires higher intensities and hence occurs at a different time along the strong-field-ionizing laser pulse envelope. In this case, what is the significance of the quoted vibrational rearrangement time?

We thank the referee for bringing up this point. In experiments with longer laser pulses and multiple initial species present, delays of an optical half-cycle or more are common due to the strong dependence of the SFI rate on the ionisation potential. Our experimental conditions, however, are different as we only have one initial species, whose depletion affects the creation of all states. Nevertheless, small delays can persist so we have adapted the main text by adding the following sentence to clarify this assignment.

“In doing so, we assume that the $\tilde{\text{E}}$ population is created at the same time as the cationic ground state. For states of the same cation, this is correct to within a few-hundred attoseconds [cite{kobayashi2018selectivity}]. This is the delay at which the population of the intermediate state is maximized and the midpoint of the CT reaction.”

Note, that we expect the 210 as delay observed between the appearance of two different double-cation species in the cited work (<https://journals.aps.org/prl/abstract/10.1103/PhysRevLett.120.233201>) to be an upper bound for the discrepancy. These states are populated through sequential double-ionisation which is a slower process than the single electron SFI which creates the B and X state.

7. The authors rightly point out that the B state has little iodine character and therefore can only be detected via transitions from the I 4d level to Rydberg states. These transitions appear in the >50 eV region. In this region, one should also find

core-to-Rydberg transitions for the other states of CF₃I⁺. However, it is unclear if these transitions are observed.

We believe the referee is referring here to the following sentence: “Instead, similar to the CF₃I neutral ground state (the spectrum of which can be found in Fig. S6), the $\tilde{\text{B}}$ -state spectral features are weaker and limited to the core-antibonding and core-Rydberg transitions which are found at spectral energies >49.5 eV (as shown in Fig. 3F).”.

While both transitions are indeed possible, we only observe the broad absorption features of core-antibonding nature. This is not unexpected as most polyatomic molecules do not exhibit clear Rydberg spectra.

We have therefore removed “and core-Rydberg” to avoid confusion.

Furthermore, strong-field ionization should also yield the A state, which is expected to undergo rapid C–I dissociation according to the potential energy curve shown in Fig. 2A. However, it seems that neither the A state nor its I* dissociation product is detected in the experiment.

The referee has rightly pointed to the peculiar absence of the A state from the experimental results. They have likely missed Section S4 of our work in which we have investigated this in depth. We have performed TDSE simulations of the vibrational dynamics along the dissociation coordinate and have decomposed the expected observables into the different polarisation components. To summarise: our current hypothesis is that the strong molecular alignment-sensitivity of SFI results in a vertically-aligned population of the A state. As the state exhibits very weak perpendicular transitions, it does not absorb the horizontally-polarised probe pulse and is therefore absent from the ATAS results. This will be investigated in future ATAS experiments on aligned CF₃I molecules but is very experimentally challenging and outside the scope of the current work.

8. The authors claim that their assignment of the 47.3 eV peak to the E state of CF₃I⁺ is “unambiguous”. As the authors point out, this transition energy agrees with the I⁺ 4d absorption at 47.32 eV (ref. 33 in the manuscript), but why does that support its assignment to the E state? Why should the XUV transition energies of the E state and the I⁺ ion be identical?

The referee brings up an interesting point - it is somewhat rare for a molecular species to have the same spectrum as an isolated ion. The similarity in the case of the E state and I⁺ is, however, supported by both the experimental results and the quantum chemistry calculations. From Fig. 3B we see that the E state absorption is far less affected by the C-I bond length than the other states and agrees very well with both the transient absorption results and the literature spectrum of I⁺. We can qualitatively understand this by looking at the electronic nature of the E state (see Fig 2B): which has two holes in the Iodine lone pairs lp1 and lp2 and, unlike the other states, one electron in the C-I anti-bonding orbital. Transitions of electrons from the core into the two lone-pair holes are responsible for the absorption between 47 and 50 eV. The interaction of these two holes (just like the two p-holes found in I⁺) lead to a richer and more intense absorption spectrum than the single-electron hole in the X states, which only exhibits spin-orbit split absorption line pairs. The

additional anti-bonding electron then provides a small amount of shielding from the CF₃ fragment, canceling approximately half of the remaining bond-dependence induced by the CI bonding electrons.

We also would like to note, that the same insensitivity to nuclear coordinates is found in ATAS on IBr (<https://www.science.org/doi/full/10.1126/science.aax0076>) where the core-to-lone-pair transitions do not shift in energy at 64.5 and 46.7 eV.

Moreover, bond dissociation along the C–I coordinate, first for the B state and subsequently for the E state, should lead to a time-dependent shift in the XUV transition energies of these states, which is not observed; instead, the transition energies of the absorption at ~54 eV and 47.3 eV remain largely unchanged as a function of time delay.

We demonstrate and explain the bond-insensitivity of the E state absorption spectrum in our previous point. For the B state, the large linewidth of the absorption spectrum complicates this analysis. While a weak red-shift of the feature can be observed, in line with the quantum chemistry calculations, it cannot be determined conclusively due to the close proximity of ground-state depletion features (<53 and >54.5 eV). As a result, we have chosen to take as large of a spectral range as possible for region IV for evaluating the transient diabatic population of the B state. By doing this, we only require transition dipole strength to remain unchanged across the CT transition, which we confirm is the case with quantum chemistry calculations in section S3.6.

Reviewer #3 (Remarks to the Author):

The manuscript *Attosecond spectroscopy of charge transfer uncovers a 1.5-fs Delay in Population Transfer* by Wörner and co-workers reports a combined theory-experiment investigation of the time delay in the electronic population transfer during the dissociation dynamics of CF_3I^+ initiated by photoionization. The dynamics is probed via attosecond transient absorption spectroscopy, allowing for a sub-femtosecond resolution and, thus, it is possible to resolve a population transfer delay of the order of 1.5 fs.

In the first part of the manuscript, the concept of population transfer delay is introduced using a quantum-dynamical picture on a one-dimensional model. The model consists of 3 electronic diabatic states $|0\rangle$, $|1\rangle$, $|2\rangle$ and is used to represent population transfer from $|0\rangle$ to $|2\rangle$. However, the states $|0\rangle$ and $|2\rangle$ are not diabatically coupled, but $|0\rangle$ is coupled to $|1\rangle$ and $|2\rangle$ is coupled to $|1\rangle$, such that population transfers from $|0\rangle$ to $|2\rangle$ via the “intruder” state $|1\rangle$.

While we understand the referees’ choice of the term “intruder state”, it is only well-defined in perturbative descriptions of problems. It might also be taken to imply that the intermediate state, on which the dynamics of the system hinge, is something that we want to get rid of from the problem at hand. This can only be done when the Schrieffer--Wolf transform is valid (see section S1.3), i.e. when the delay is absent, and misses an important point of our work: that the population transfer delay could be quite ubiquitous and may require us to more-often consider the effect of higher- or lower-lying states when inferring (photo-)chemical reaction pathways.

Such transfer is delayed with respect to a direct transfer from $|0\rangle$ to $|2\rangle$. Such a delay is estimated of the order of 1 to 2 fs using typical molecular valence state parameters in the model. In the second part of the manuscript, the electronic structure of CF_3I^+ is analyzed using the ab-initio multiconfigurational second-order perturbation theory restricted active space (RASPT2) method, and the experimental signals, namely the transient absorption spectra, are interpreted based on the RASPT2 calculations. In the third part of the manuscript, the transient absorption signal is interpreted based on the model introduced in the first part.

While we believe that experiments with such a temporal resolution are able to highlight *some* time delay during the charge transfer process occurring upon photoionization, we are not convinced that it can be interpreted as done by the authors, namely by hypothesizing the presence of an “intruder” electronic state which neither is observed in experiments nor appears in the calculations. Therefore, at this stage, we cannot recommend publication of the manuscript in Nature Communications and we invite the authors to address the following points.

1. The interpretation of the time delay in the population transfer between B and E in terms of the “intrusion” of an additional state I that cannot be observed, by the quantum chemistry calculations and by experiments, is confusing and perhaps not even necessary. The analysis of the occupation of the molecular

orbitals and the single electron picture used to describe the change of electronic character (Fig.2) suggest a two-electron rearrangement process that is certainly not instantaneous. Can it be that the authors observe this delay in their experiment?

We believe that a slight confusion of different representations is obfuscating the fact that we actually all agree. The problem at hand can be viewed both as the interaction of different electronic states and as the rearrangement of electrons in orbitals. When using the adiabatic or diabatic state representation, we are not able to speak of the motion of individual electrons but the total population and energy are easily conserved. As a result, population that is lost from the B state that is not immediately observed in the E state must be in some other state, which we have called I. When we use the orbital representation, electronic rearrangement is more intuitive but it becomes more difficult to properly define a state, its population, and apply energy conservation.

The referee's statement in the orbital picture that the observed delay is due to "a two-electron rearrangement process that is certainly not instantaneous" is equivalent to stating that the reaction passes through an intermediate state in the diabatic state representation.

This connection is implied in the caption of Figure 2 and previous drafts of the manuscript discussed it more explicitly, however, this was removed for the sake of brevity and to make the work appeal to a broader audience. In response to this comment we have modified the following sentence:

"The electronic characters, meanwhile, indicate that the studied CT requires the simultaneous rearrangement of two electrons as illustrated by dashed-dotted arrows in Fig.~2B."

to:

"A transition between these states requires the rearrangement of two electrons (see Fig.~2B) and is symmetry-forbidden in the case of direct vibronic coupling, suggesting that an intermediate electronic state must be involved in the CT reaction, i.e. the two-electron rearrangement cannot occur instantaneously."

We would also like to add that it may be possible to apply second quantisation to this problem to provide a formal but intuitive explanation for the population delay as suggested by the reviewer. Such an approach has been successful for treating autoionizing 2-electron arrangements such as inter-atomic/molecular coulombic decay (<https://doi.org/10.1103/PhysRevB.64.245104>) but its application to a bound-bound vibronic transition has so-far remained limited (<https://doi.org/10.1063/5.0018930>). We hope this work can stimulate further research in this direction.

2. As follows from point 1., the dynamical picture used to introduce the concept of population transfer delay and to validate the experimental observations is not convincing. If the authors strongly believe that this interpretation is, instead, robust, we invite them to find the state(s) *I* at least with quantum chemistry calculations.

In section S5, we use the results of the quantum chemistry calculations together with symmetry arguments to determine the electronic configuration and total symmetry of the intermediate state. These configurations are $(|p_{3\sigma_4}^1 p_{\{1,2\}\sigma_5}^3\rangle$ and $(|p_{3\sigma_4}^2 p_{\{1,2\}\sigma_5}^3\rangle$ giving an E symmetry to the intermediate state. However, as we mention on page 11 and

discuss in more detail in section S5, the coupling of the B, E and I states is symmetry forbidden in the C_{3v} point group. This lowers the degree of configuration mixing and prevents the high-energy doubly-excited intermediate configuration from contributing to the lower energy states accessible in our calculations; and therefore the direct identification of the intermediate state from the RASPT2 calculations. Meanwhile, we do observe a small amount of mixing of the ground state configuration into the highly mixed $lp1$, $\sigma4$, $\sigma5$ state (the 9th state at the conical intersection) but not to a degree that we can be confident to assign it as the intermediate state.

We believe that more computationally-efficient multireference ab-initio calculations (which include spin-orbit coupling) performed at distorted geometries might be able to directly identify the intermediate state, however, these are outside of our current capabilities and the scope of the present work. Our hope is that this work might stimulate such further studies.

These arguments have been added to Section S3.5 of the SM, supplemented by a more detailed analysis of the calculations for the lowest 10 electronic states at C-I bond lengths around the state crossing.

3. While the analysis of the couplings between B, E and I based on symmetry considerations holds valid at the C_{3v} equilibrium geometry along the elongation of the C-I bond, the ab initio dynamical picture of the molecule is quite different, and symmetry is probably broken during the dynamics. Therefore, distorted geometries should be investigated.

The vibronic-coupling analysis provided in the SM, section S5, has been formulated in the C_{3v} point group. This does not imply, however, that it is only valid for C_{3v} geometries. This is a well-established, although rarely mentioned, fact in molecular group theory. CF3I has a complete nuclear permutation inversion (CNPI) group given by the direct product of the S_3 permutation group and the inversion group: $S_3 \times \{E, E^*\}$. (For details, see Bunker&Jensen, Molecular Symmetry and Spectroscopy, National Research Council of Canada Research Press, 1998). The CNPI group contains a number of operations that possess a very large energy barrier (e.g. the inversion of the molecule). These energy barriers exceed the kinetic energy available for molecular dynamics in our experiments, such that they correspond to unfeasible operations. Keeping only the feasible operations leaves us with the operations of the $C_{3v}(M)$ group, the character table of which is reproduced below (taken from the above-mentioned book). The molecular symmetry group $C_{3v}(M)$ is isomorphic with the C_{3v} point group. Therefore, the symmetry arguments formulated for the C_{3v} point group in Section S5 of the SM also apply to the $C_{3v}(M)$ group, which describes CF3I in all nuclear configurations (geometries) accessible under our experimental conditions, not only the C_{3v} -symmetric ones. As a consequence,

the vibronic coupling arguments remain unchanged when considering distorted geometries and working with the molecular-symmetry group $C_{3v}(M)$.

672

Table A-6
The group $C_{3v}(M)$
Example: Methyl fluoride

$C_{3v}(M)$:	E	(123)	$(23)^*$	
	1	2	3	
C_{3v} :	E	$2C_3$	$3\sigma_v$	
Equiv. rot.:	R^0	$R_z^{2\pi/3}$	$R_{\pi/2}^\pi$	
A_1 :	1	1	1	$T_z, \alpha_{zz}, \alpha_{xx} + \alpha_{yy}$
A_2 :	1	1	-1	\hat{J}_z, Γ^*
E :	2	-1	0	$(T_x, T_y), (\hat{J}_x, \hat{J}_y), (\alpha_{xz}, \alpha_{yz}), (\alpha_{xx} - \alpha_{yy}, \alpha_{xy})$

This explanation has been added to the SM, Section 5.

We hope that this explains why our symmetry analysis is valid. The referee is, however, correct that the distorted geometries are very important to the dynamics. As we have noted in Section S5 the population transfer happens at distorted geometries slightly displaced from the conical intersection where the RASPT2 calculations are performed and it is exactly at these geometries that our single-effective-dimension 3-state model describes.

- The results presented in Fig.4 are not convincing. First of all, in the upper panel of B the authors do not explain how the observed transient absorptions (reported in A) are “rescaled” to fit into the diabatic populations dynamical picture – which is the picture used to introduce the concept of population transfer delay (Fig.1) in the manuscript. While in Fig.1 the presence of non-zero population P1 at time 0 is acceptable since it is just a theoretical model, in panel B of Fig.4 the authors need to explain how the state I is populated by the excitation. It seems to us that the dynamics is initialized by fully populating the adiabatic ground state (dashed blue curve in C bottom panel), which then has contributions in the 3 diabatic states. However, this is somehow counterintuitive when comparing theory with experiments because the bright electronic states in the Franck-Condon region are usually identified based on symmetry arguments on the diabatic states.

We believe that the reviewers are referring to Fig. 5 instead of Fig. 4 here. Assuming this to be the case, we can comment on this as follows:

- In the second paragraph on page 4 we explicitly state that the population delay “ is extracted from the difference of the position parameters μ of the fits”. The rescaling that is performed in panel B therefore has no influence on the timescales we report or our conclusions. In the same paragraph we explain how the rescaling/renormalisation procedure is performed and that it is only there to help the reader see the delay visually. We want to underline that no information is extracted from the rescaling procedure.
- Note that this makes the population transfer delay a very useful observable in experimental studies as it can be directly extracted from measurements without the need for any input from quantum chemistry calculations.
- The final paragraph on page 6 explains why we can consider the ionic states to be initialized in the adiabatic ground state; namely that the SFI process with IR fields in small molecules proceeds electronically-adiabatically.
- Although for geometries with C_{3v} symmetry (as is the case in our quantum chemistry calculations) the B state is symmetry-forbidden from mixing with the I state, this is not the case for distorted geometries. As the CT reaction in CF₃I occurs at these geometries (see section S5), we can take the I state to also be populated in the experiment.
- Finally, it is important to note that population transfer delay is present in our TDSE simulations whether the system is initialised in diabatic or adiabatic ground state. An adiabatic initialization simply suppresses beats in the diabatic population that are also observed in simulations of the LZSM crossing simplifying the fitting procedure and therefore the extraction of the parameters. The results of such an “initially diabatic” simulation are shown below. The figure shows the total population (black, the diabatic populations (light blue = initial, orange = intermediate, green = final) and the AGDN fits (blue and red).

- The delay extracted from this ‘diabatic initialization’ is only 12 attoseconds shorter than in the ‘adiabatic’ case. We are therefore confident that neither the renormalisation, nor the initialisation of the system in the adiabatic vs diabatic ground state has an effect on the conclusions of this work

5. The population transfer delay is referred to as an *observable* in the manuscript. However, it can be deduced only in the diabatic picture while it seems to disappear in the adiabatic picture, as the population indicated in dashed blue in Fig.4B upper panel is always equal to 1 since the process is fully adiabatic. How do the authors justify the fact that this *observable* depends on the representation?

We thank the referees for bringing up this point and helping us clarify this in the manuscript. It is true that the delay between loss/appearance of diabatic populations cannot be directly associated with a single quantum mechanical observable operator. In our experiment, access to this observable is provided through the transition dipole operator. Particularly in the case of core-level absorption spectroscopy, the transition dipole is very sensitive to the electronic configuration (i.e. electronic character) of the system because the transitions are predominantly single-electron rearrangements. This is verified through our cutting edge RASPT2 calculations which show no dependence of the transition dipole magnitude on the C-I bond length (see section S3.6) and allow the diabatic populations (in a representation that preserves the electronic configuration of the states with respect to the C-I bond length) to be set proportional to the transient absorption signal.

To clarify this point we have adapted the first paragraph from of S3.6 and inserted it at the bottom of page 10/start of page 11:

“To make the direct connection between the experimental absorption observable and diabatic populations (and eventually a delay), we must establish the invariance of the absorption cross section with respect to C-I coordinate in the vicinity of the CT crossing. This is confirmed in section S3.6 by spectrally integrating the RASPT2 results in and vicinity of the crossing, confirming that regions II and IV can be used as measures of the transient diabatic populations of the $\tilde{\text{B}}$ and $\tilde{\text{E}}$ states.”

The delay is an observable because it can be observed in our experiment etc.

Minor points:

6. An explanation of the reason why in Fig.3F (lower panel) the C-I bond length of 2.34 Angstrom is chosen to compute the spectrum should be provided.

The exact choice of this bond length is not critical to the analysis - as we can see from Fig. 4 and Fig. S12, the spectra of the X and E states do not vary much in the range of 2.2 and 2.5Å. The chosen bond length was chosen as it was approximately the middle of this range.

7. Can the authors justify more clearly how the transient absorption of states B and E reported in Fig.4A gives direct access to the populations of these states? The fact that their initial values are basically the same seems to not support the interpretation as one state being not populated at the beginning.

To answer this question we turn to Fig. 3F. Here, both the experimental and theoretical results show that the absorption of the X states extend somewhat into the spectral region that we integrate to evaluate the population of the E state. This leads this spectral region to show an initial rise in absorption at the same time as the X states. As seen in figure 4B, however, this contribution is stable for the first 30 fs allowing it to be safely neglected. Here we see again how the population transfer delay is a robust observable that can be easily extracted in the presence of other signals.

Reviewer #4 (Remarks to the Author):

We thank reviewer #4 for their contribution to the peer review of our manuscript and particularly appreciate their input and recommendations that have helped us to improve our manuscript.

We hope that our replies are convincing to reviewer #3 and #4, and that we have been able to show that our treatment of the problem is in-line with their correct intuition that a 2-electron rearrangement is a process that should not be able to occur instantaneously.

Response to the referees

We would like to thank the referees for their valuable time they have taken to review our manuscript and their insightful comments and for their excellent suggestions, which helped us to further improve our manuscript. Below, we have color coded each referee's comment in blue, our replies in black and additions/changes in green in our resubmission. We hope that the referees will find our answers and the revised manuscript satisfactory.

Reviewer #1 (Remarks to the Author):

I considered this paper as excellent work already in my previous review. My minor points have been properly addressed. In addition I had a look at the comments of the other reviewers and the related response by the authors. In my view, the authors properly addressed the points and clarified some misunderstandings. Of course, there could be room for improvement of the theoretical model, but this would go beyond what's needed to support the main conclusions. I recommend publication.

We thank the reviewer for approving our revisions and for recommending publication.

Reviewer #2 (Remarks to the Author):

I greatly appreciate the authors' detailed replies to the reviewers' comments. In most cases, the replies are satisfactory. However, the identity of the intermediate state remains unclear. While they have undertaken a symmetry analysis to show that the intermediate state has 2E symmetry, they have yet to identify the state. Maybe another way to phrase my question is to ask which of the potential curves in the lower panel of Fig. 4B is that of the intermediate state, or which of the 2E states in Table S2 is the intermediate state?

We are grateful to the reviewer for insisting on the identification of the intermediate state. We are very happy to report that, contrary to our initial expectations, we have managed to obtain converged calculations of the intermediate state.

These results have been added to Sections S3.2 and S3.5 and in the new table S5. They have been mentioned in the main text (p. 6): "A transition between these states requires the rearrangement of two electrons (see Fig. 2B), suggesting that it cannot occur instantaneously and that an intermediate electronic state I of mixed $(|p_3\rangle^1(\sigma_4)^2(|p_{1,2}\rangle^3(\sigma_5^*)^1)$ character might be involved. We confirm the existence of such a state ~4 eV above the CT crossing using the density functional theory/multireference configuration interaction (DFT/MRCI) method²⁴⁻²⁶, a more computationally efficient multireference approach (see Sections S3.2 and S3.5 for details)."

Reviewer #3 (Remarks to the Author):

We thank the authors for their answers to the reviewers and for their clarifications, especially in the response letter. Indeed, we still believe that theoretical validation of the

existence of the state(s) I based on quantum chemistry would make the observations reported in this work more robust (and it does not seem at all beyond the scope of the present work). However, we appreciate the originality of the experiment and of its analysis reported in the manuscript and we believe that it has indeed the potential to stimulate further studies. Therefore, we can recommend the manuscript for publication in Nature Communications.

We are grateful to reviewer for recommending publication of our manuscript and the identification of the intermediate state. We are very happy to report that, contrary to our initial expectations, we have managed to obtain converged calculations of the intermediate state.

These results have been added to Sections S3.2 and S3.5 and in the new table S5. They have been mentioned in the main text (p. 6): "A transition between these states requires the rearrangement of two electrons (see Fig. 2B), suggesting that it cannot occur instantaneously and that an intermediate electronic state I of mixed $(lp_3)^1(\sigma_4)^2(lp_{1,2})^3(\sigma_5^*)^1$ character might be involved. We confirm the existence of such a state ~ 4 eV above the CT crossing using the density functional theory/multireference configuration interaction (DFT/MRCI) method²⁴⁻²⁶, a more computationally efficient multireference approach (see Sections S3.2 and S3.5 for details)."

Reviewer #4 (Remarks to the Author):

We thank the reviewer for their valuable input to the review of this manuscript.